# Two Different Strains of Severe Fever with Thrombocytopenia Syndrome Virus (SFTSV) in North and South Osaka by Phylogenetic Analysis of Evolutionary Lineage: Evidence for Independent SFTSV Transmission

**DOI:** 10.3390/v13020177

**Published:** 2021-01-25

**Authors:** Ryo Ikemori, Ikuko Aoyama, Tadahiro Sasaki, Hirono Takabayashi, Kazutoshi Morisada, Masaru Kinoshita, Kazuyoshi Ikuta, Takahiro Yumisashi, Kazushi Motomura

**Affiliations:** 1Virology Section, Division of Microbiology, Osaka Institute of Public Health, Osaka 537-0025, Japan; ikemori@iph.osaka.jp (R.I.); aoyama@iph.osaka.jp (I.A.); sasatada@biken.osaka-u.ac.jp (T.S.); ikuta@iph.osaka.jp (K.I.); yumisasi@iph.osaka.jp (T.Y.); 2Research Institute of Microbial Diseases, Osaka University, Suita, Osaka 565-0781, Japan; 3Fujiidera Public Health Center, Fujiidera, Osaka 583-0024, Japan; TakabayashiH@mbox.pref.osaka.lg.jp; 4Ikeda Public Health Center, Ikeda, Osaka 563-0041, Japan; 5Takatsuki-City Public Health Center, Takatsuki, Osaka 569-0052, Japan; morisada-k@city.takatsuki.osaka.jp; 6Department of Health and Medical Care, Osaka Prefectural Government, Osaka 540-8570, Japan; KinoshitaMas@mbox.pref.osaka.lg.jp

**Keywords:** SFTS, genotype, whole genome

## Abstract

Severe fever with thrombocytopenia syndrome (SFTS) is a novel tick-borne infectious disease, therefore, the information on the whole genome of the SFTS virus (SFTSV) is still limited. This study demonstrates a nearly whole genome of the SFTSV identified in Osaka in 2017 and 2018 by next-generation sequencing (NGS). The evolutionary lineage of two genotypes, C5 and J1, was identified in Osaka. The first case in Osaka belongs to suspect reassortment (L:C5, M:C5, S:C4), the other is genotype J1 (L: J1, M: J1, S: J1) according to the classification by a Japanese group. C5 was identified in China, indicating that C5 identified in this study may be transmitted by birds between China and Japan. This study revealed that different SFTSV genotypes were distributed in two local areas, suggesting the separate or focal transmission patterns in Osaka.

## 1. Introduction

Severe fever with thrombocytopenia syndrome (SFTS) is a novel tick-borne infectious disease with high mortality. The first reported cases of SFTS were reported in China in 2011 [1], and later the virus was identified in other countries in Asia such as Korea, Japan, Vietnam, Taiwan, and Myanmar [2,3,4,5,6].

SFTS virus (SFTSV), recently renamed *Dabie bandavirus*, belongs to the family Phenuiviridae, being an enveloped virus containing a three-segment negative RNA genome [1,7]. SFTSV is most closely related to Guertu virus and Heartland virus, which are tick-borne viruses like SFTSV, in the family Phenuiviridae [8]. The three segments are referred to as large (L), medium (M), and small (S) in order from the longest. The L segment encodes RNA-dependent RNA polymerase (RdRp), the M segment encodes two glycoproteins (GP), Gn and Gc, and the S segment encodes nucleoprotein (Np) and nonstructural protein (NS).

SFTS was identified for the first time in 2011; therefore, the genotyping has not been established yet in the world. Yoshikawa et al. classified SFTSV into eight genotypes: Chinese clades 1 to 5 (C1 to C5 clades) and Japanese clades 1 to 3 (J1 to J3 clades) [9]. These previous studies revealed that the distribution of SFTSV genotypes has regional characteristics. J1–J3 clades were mainly detected in Japan and Korea, whereas C1–C5 clades were found in China [9,10]. In Korea, it is estimated that 70% of the SFTSV belong to the J clade; in addition, recently, the C clade has also been identified there [9,11]. The C clade has been detected in Japan, but there are fewer reports than in Korea [9]. There are only few whole genome analyses limited to classifying genotypes. Some previous studies analyzed only a partial region or only the M or S segments; therefore, there are few complete SFTSV sequences in the public database. SFTSV showed different genotypes in each segment, indicating that SFTSV undergoes reassortment like the influenza virus and the rotavirus for genome segments [10,12].

We confirmed two cases in Osaka. The first case was found in 2017 and the other was found in 2018. Two confirmed cases were determined by RT-PCR. To investigate the genetic background of the SFTSV circulating in Osaka, we performed viral isolation, genome analysis using next-generation sequencing (NGS), and phylogenetic analysis.

## 2. Materials and Methods

### 2.1. Virus Isolation

The patient serum of the first case was collected in August 2017 and the second case’s serum was collected in July 2018. The specimen was inoculated with Vero cells in an incubator during one week for the isolation of the SFTSV. Vero cells were had been observed to detect the cytopathic effect (CPE) every day for one week. The supernatant was collected at −80 °C and used for extraction of viral RNA. 

### 2.2. Whole Genome Sequencing by NGS

The viral RNA was extracted from the supernatant by using a QIAamp Viral RNA kit (Qiagen, Hilden, Germany) and stored at −80 °C. The cDNA was used for high-throughput sequencing using a Nextera XT DNA Sample Prep Kit (Illumina, San Diego, CA, USA). Sequence-specific adapters were ligated to the amplicons and then the fragments were amplified using seven cycles of the polymerase chain reaction. The sequencing was performed on an automated MiSeq (Illumina) after the preparation of DNA libraries for MiSeq. We used CLC Workbench (Filgen, Nagoya, Japan) to construct a nearly whole genome and assemble the short reads.

### 2.3. Phylogenetic Analysis

The maximum likelihood tree of the nucleotide sequences was constructed. Briefly, the sequences were aligned using CLUSTAL W (version 1.4) and a distance matrix of nucleotide substitutions per site was estimated from the alignment according to Kimura’s two-parameter method. The trees were generated with 100 bootstrap replicates from the matrix numbers using MEGA version 7.0 [13] (https://www.megasoftware.net/).

## 3. Results and Discussion

Firstly, we investigated the phylogeny of the three fragments of the SFTSV’s near-full-length genome sequences including segment L, segment M, and segment S (about 10.0 kb). For this study, we used the sequences obtained in Osaka (n = 2), 78 reference sequences of a past global or Japanese epidemic reported by various studies, and various reported sequences of other genotypes (Appendix A). Case A (strain name: JaOF2017-7 ) was found in the south of Osaka, the putative exposure area was the south of Osaka or Wakayama in 2017. Case B (strain name: JaOT2018-6 ) was found in the north of Osaka, the putative exposure area was Shiga in 2018. Figure 1 shows a representative maximum likelihood tree constructed by using 78 reference sequences. The tree show that that the two newly obtained sequences are divisible into two distinct lineage groups with the high bootstrap value of 100/100 according to the classification by a Japanese group (Figure 1) [9]. The segment L of case A belonged to C5 and that of case B belonged to J1 (Figure 1A). The phylogenetic tree of segment M revealed that case A belonged to C5 and case B belonged to J1 (Figure 1B). The phylogenetic tree of segment S revealed that case A belonged to C4 and case B belonged to J1 (Figure 1C). The monophyletic relationship of the sequences was reproducible when the trees were constructed with different algorithms, such as neighbor joining and Unweighted Pair Group Method with Arithmetic mean (UPMGA), or with the segment L, segment M, or segment S sequences (data not shown). According to the phylogenetic tree analysis, case B formed a monophyletic group with the J1 references, which was the major dominant genotype in Japan. On the other hand, segments of case A formed a monophyletic group with the C5 references which were reported in China [9] and were closely related to the SPL161A isolated in Wakayama. In Japan, mainly, J1–J3 strains were detected, and C1–C5 genotypes were barely reported.

In this study, we examined a nearly whole genome of the SFTSV identified in Osaka in 2017 and 2018 by NGS. We analyzed the evolutionary lineage of two genotypes, C5 and J1, in the north and the south of Osaka located at the distance of about 60 km. We first showed the nearly whole genome of the C5 identified in Osaka. Case A was found in the south of Osaka, the putative exposure area was Wakayama. The sequence of segments L, M, and S in case A shows high homology (99%) with that of the SPL161A isolated in Wakayama, indicating that C5 was cocirculating in Osaka and the Wakayama area where it had a possibility of spread and persisted. The presence of C5 may be related to the area near the border in Wakayama where C5 was reported recently, suggesting that C5 is circulating and spread in the Wakayama area. C5 is found in China and is less often reported in Japan. It is hard to think that humans (e.g., tourists) are able to carry SFTSV, because SFTS induces severe symptoms in humans. The birds are able to carry SFTS virus-bearing ticks for a long distance, indicating that birds between China and Japan may transmit C5 [10,14,15].

Interestingly, segments L and M were clustered in C5, but segment S was clustered in C4, indicating that this case A would be a possible reassortment strain, because it has already been reported that SPL161A which shows high homology (99%) with case A would be a possible reassortment strain [9]. There are few sequences of C5 genotypes in the public database; therefore, we are not able to confirm the conclusion that case A is a reassortment strain. According to the classification by a Japanese group, segments L and M were defined for C1–C5, but segment S was classified only for C1–C4 [9]. In addition, the classification of genotypes is still controversial depending on the country [11,16]. It is reported that bunyavirus sometimes occurred as a reassortment virus by changing segment M, it may emerge as a new reassortment strain of SFTSV [17]. However, the genotype pattern of case A revealed that only segment S was changed, not segment M. It may have been the result of accumulation of nucleotide variability, and case A may apparently show a reassortment strain between the genotypes.

In summary, we demonstrated distribution of different SFTSV genotypes in two local areas in Osaka, suggesting separate or focal transmission patterns in Osaka.

## Figures and Tables

**Figure 1 viruses-13-00177-f001:**
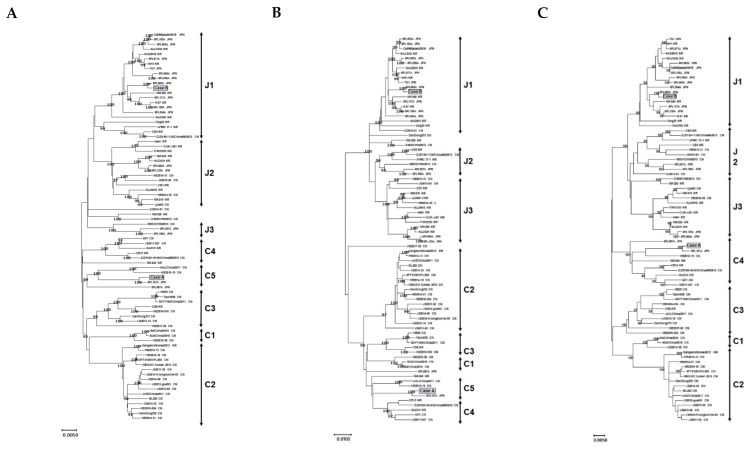
Phylogenetic classification of the SFTSV. The maximum likelihood tree was constructed with the near-full-length genome sequences (about 10.0 kb) obtained from the specimens collected in Osaka in 2017 and 2018 in this study (*n* = 2), SFTSV reference sequences from past epidemics in Japan and other countries (*n* = 78). The phylogenetic trees are shown as segment L (**A**), segment M (**B**), and segment S (**C**). The bootstrap value is shown to be above 90.

## Data Availability

Data available in a publicly accessible repository; The data presented in this study are openly available in GenBank, accession number [LC590890-LC590895].

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
