# Peer review of "Two Different Strains of Severe Fever with Thrombocytopenia Syndrome Virus (SFTSV) in North and South Osaka by Phylogenetic Analysis of Evolutionary Lineage: Evidence for Independent SFTSV Transmission"

_viruses, 2021, doi:10.3390/v13020177_

Round 1
Reviewer 1 Report
The manuscript describes nearly full genomes of two new variants of SFTSV from patients. SFTS is an emerging infection and new information is needed to understand the risks associated with this pathogen. Patient-derived sequences are rare and hence the manuscript has significant novelty. However, I feel that the manuscript can be much revised starting from an extensive editing of the English language.
In addition, data on the patients should be added describing their symptoms and disease progression. At what time after onset of symptoms were the samples collected?
The phylogenetic tree should be made more informative by adding information on from where the sequences are originating (patients, ticks, other species?).
Finally, do describe briefly the updates to the taxonomy and what are the closest relatives in the family Phenuiviridae.
Author Response
Responses to the Reviewer Comments
Reviewer 1:
Comments to the Author:
The manuscript describes nearly full genomes of two new variants of SFTSV from patients. SFTS is an emerging infection and new information is needed to understand the risks associated with this pathogen. Patient-derived sequences are rare and hence the manuscript has significant novelty. However, I feel that the manuscript can be much revised starting from an extensive editing of the English language.
Response: Thank you very much for such important insights. We are delighted to hear that you found our work meaningful in this field. In the following sections, you will find our responses to each of your comments. We are grateful to you for reviewing our manuscript. I am sorry for our poor English writing, but this manuscript was revised by commercial editing company before submission. We claimed it to the company.
- In addition, data on the patients should be added describing their symptoms and disease progression. At what time after onset of symptoms were the samples collected?
Response: Thank you for your comment. We are not able to approach the information of patients for the privacy policy. Unfortunately, we could not describe the symptoms, disease progression in detail.
- The phylogenetic tree should be made more informative by adding information on from where the sequences are originating (patients, ticks, other species?).
Response: Thank you for the suggestion. We described the originating of sequences in supplementary table.
- Finally, do describe briefly the updates to the taxonomy and what are the closest relatives in the family Phenuiviridae.
Response: We appreciate your comment. We confirmed that SFTSV was renamed Dabie bandavirus following the updated taxonomy. In addition, SFTSV is most closely related to Guertu virus and Heartland virus in the family Phenuiviridae. According to the comments, we changed the sentence and put new reference as below (Line 41-44);
“ SFTS virus (SFTSV), recently renamed Dabie bandavirus, belongs to the family Phenuiviridae, which is an enveloped virus containing a three-segment negative RNA genome [1,7]. SFTSV is most closely related to Guertu virus and Heartland virus, which are tick-borne viruses same SFTSV, in the family Phenuiviridae [8].”
Reviewer 2 Report
The title should be changed to read 'Two different' instead of 'Different two'
The methods for viral isolation should be expanded a little to be more descriptive.
Author Response
Responses to the Reviewer Comments
Reviewer 2:
Comments to the Author:
- The title should be changed to read 'Two different' instead of 'Different two'.
Response: Thank you for these important insights. We changed the title to “Two different” following the comment.
- The methods for viral isolation should be expanded a little to be more descriptive.
Response: Thank you for this comment. We have added sentence as below (Line 67– Line 68).
“The Vero cells was observed to detect cytopathic effect (CPE) every day for one week.”
Reviewer 3 Report
Major
Authors has demonstrated two full genome of SFTSV identified in Osaka in 2017 and 2018 by new-generation sequence (NGS), and identified that the evolutionary lineage of two genotypes, C5 and J1.
1. Are these two full genome represent in Osaka? Because I worry that two genome is not enough for phylogenetic analysis. So, could you more detail about this?
2. Author showed that one is Japan clade “J1” (L: J1, M: J1, S: J1) and the other is China clade “C5” (L:C5, M:C5, S:C4).
Is China clade first reported in Japan? If china clade is already found (or existed) in Japan, please explain the difference between previous found and your finding.
3. Did you also compare with Korea clade? Author have to compare with Korea clade, because Korea is located among China, Korea, and Japan and this clade “C5” could be transmit via Korea.
4. In line 59 and 60, please put detail information of these isolated virus and patients.
5. In line 112, authors wrote that “indicating that birds between China and Japan may transmit C5” Please more detail write the reason and the role of migratory birds and reference (s) about migratory birds are long distance carrier about SFTS virus-bearing tick.
6. In line 114, authors wrote that “this case A would be a possible re-assortment strain”.
Could you more detail write about this issue (re-assortment) and put reference (s) for SFTSV re-assortment?
Minor
1. In line 36, please recheck first reported year (2011) of SFTS in China.
2. In line 37, SFTS also reported in other countries in Asia such as Vietnam, Taiwan, and Myanmar. Please put these and references in line 37.
3. In line 40, Please change “short” to “small”.
4. In line 49 and 50, please put reference (s) in this issue.
5. In line 81 and 82, case A is isolated from the south of Osaka and B is isolated from the north of Osaka.
Could you more detail write the geographic difference between south and north?
Author Response
Responses to the Reviewer Comments
Reviewer 3:
Comments to the Author:
- Authors has demonstrated two full genome of SFTSV identified in Osaka in 2017 and 2018 by new-generation sequence (NGS), and identified that the evolutionary lineage of two genotypes, C5 and J1. Are these two full genome represent in Osaka? Because I worry that two genome is not enough for phylogenetic analysis. So, could you more detail about this?
Response: Thank you for these important insights. We are delighted to know that you think our work is meaningful. In the following sections, you will find responses to each of your comments. We are grateful to you for reviewing our manuscript. Only two SFTS cases were detected in Osaka, therefore, we analyzed phylogenic analysis for two genome of SFTSV, C5 and J1.
- Author showed that one is Japan clade “J1” (L: J1, M: J1, S: J1) and the other is China clade “C5” (L:C5, M:C5, S:C4). Is China clade first reported in Japan? If china clade is already found (or existed) in Japan, please explain the difference between previous found and your finding.
Response: Thank you for your comment. Some C clades were detected in Japan, especially C5 is closely related and high homology (99%) with SPL161A identified in Wakayama province where is south of Osaka province. Although nearly full genome of L segment and S segment in SPL161A were analyzed and opened in public database (accession number; AB983534.1 and AB985572.1), half length of M segment in SPL161A was analyzed (accession number; AB985657.1). Our study is first paper that nearly full genome of L segment, M segment and S segment in C5 clade were analyzed.
- Did you also compare with Korea clade? Author have to compare with Korea clade, because Korea is located among China, Korea, and Japan and this clade “C5” could be transmit via Korea.
Response: Thank you for these insights. According to the classification of genotypes by Japanese group, most of SFTSV identified in Korea belong to J clade, therefore, we think that “compare with Korea clade” means “compare with Japanese clade”. In addition, J clade might be brought from China via South Korea by migratory birds. We described this point in response for question 5. On the other hand, there is only one case of C5 clade is identified in Japan, therefore, it is hard to tell how the C5 clade is brought to Japan. We think that it is high possibility to be carried by migratory birds same as J clades. In order to describe more information about the distribution of genotypes in Korea, we modified the sentence and put new reference as below (Line 51- Line 54);
“J1–J3 clades were mainly detected in Japan and Korea, whereas C1–C5 clades were found in China [9,10]. In Korea, estimated 70% of SFTSV belong J clade, in addition, recently C clade is also identified [9,11]. C clade has been detected in Japan, but there are fewer reports than that of Korea [9].”
- In line 59 and 60, please put detail information of these isolated virus and patients.
Response: Thank you for your comment. We are not able to approach the information of patients for the privacy policy. Unfortunately we could not describe the symptoms, disease progression in detail.
- In line 112, authors wrote that “indicating that birds between China and Japan may transmit C5” Please more detail write the reason and the role of migratory birds and reference(s) about migratory birds are long distance carrier about SFTS virus-bearing tick.
Response: Thank you for these insights. Chinese (C) clade are mainly detected in China, on the other hand, it is less cases in Japan. It is hard to think that human (ex. tourists) is able to carry SFTSV, because SFTS brings severe symptoms for human. SFTSV is transmitted by tick, and the birds is able to carry SFTS virus-bearing tick for long distance. Two references suggest this working hypothesis, therefore, we have added references suggesting that SFTSV transmit from migratory birds (reference number 10, 14 and 15). We have added sentences as below (Line 132– Line 135).
“ C5 is mainly found in China and less reported in Japan. It is hard to think that human (ex. tourists) is able to carry SFTSV, because SFTS brings severe symptoms for human. The birds is able to carry SFTS virus-bearing tick for long distance, indicating that birds between China and Japan may transmit C5[10,14,15].”
- In line 114, authors wrote that “this case A would be a possible re-assortment strain”. Could you more detail write about this issue (re-assortment) and put reference (s) for SFTSV re-assortment?
Response: Thank you for these insights. C5 identified in case A is high homology with SPL161A. It is already reported that SPL161A would be a possible re-assortment strain with reference to Yoshikawa et al (reference No. 9). In addtion, SFTSV re-assortment emerges with reference to Shi et al and Yun et al (reference No. 10,11). We have added sentences as below (Line 137–139) ;
“ Because it is already reported that SPL161A which is high homology (99%) with case A would be a possible re-assortment strain [9].”
Minor comments:
- In line 36, please recheck first reported year (2011) of SFTS in China.
Response: Thank you for your comment. We confirmed the reference No.1 that the authors identified SFTSV from clinical sample collected in 2009. We changed the sentence as below (Line 39) ;
“ The first reported year (2011) of SFTS was published in China”
- In line 37, SFTS also reported in other countries in Asia such as Vietnam, Taiwan, and Myanmar. Please put these and references in line 37.
Response: Thank you for your comment. I am sorry that I forgot these references in your point.
Vietnam case:(https://pubmed.ncbi.nlm.nih.gov/31002059/)reference No.4
Taiwan case:(https://pubmed.ncbi.nlm.nih.gov/32568054/)reference No.5
Myanmar case:(https://www.ncbi.nlm.nih.gov/pmc/articles/PMC7392420/)reference No.6
We changed the sentence and added three references as below (Line 39-41);
“ later the virus was identified in other countries in Asia such as Korea, Japan, Vietnam, Taiwan, and Myanmar [2–6].”
- In line 40, Please change “short” to “small”.
Response: Thank you for your comment. We changed “Small” (Line 45).
- In line 49 and 50, please put reference (s) in this issue.
Response: Thank you for your comment. We put reference number 9, 10 and 11.
- In line 81 and 82, case A is isolated from the south of Osaka and B is isolated from the north of Osaka. Could you more detail write the geographic difference between south and north?
Response: Thank you for your comment. These patients were infected in another place not Osaka province, therefore, it might be no meaning to describe the geographic difference in detail. Unfortunately we are not able to identify the confirmed infected area. We put the sentence as below (Line 125-126);
“ in the north and the south of Osaka where locates the distance about 60km”.
Round 2
Reviewer 3 Report
Nothing